# What Happened to the 'Truth Universally Acknowledged'? Translation as Reception of Jane Austen in France

Isabelle Bour

Department of the English-Speaking World, Université Sorbonne Nouvelle, 75005 Paris, France; isabelle.bour@sorbonne-nouvelle.fr

**Abstract:** There are now, in 2022, sixteen French translations of Jane Austen's *Pride and Prejudice*. The *incipit* includes one of the most famous statements in the English language, as well as a modal auxiliary, the rendering of which constitutes a minor challenge for any translator. This essay will analyse all translations of the *incipit*, relating translation choices to historical circumstances, the contemporary status of British literature and attitudes to the translation of fiction as well as to the state of the book market.

**Keywords:** opening; incipit; Jane Austen; translations; pride and prejudice

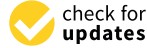

The starting-point for this essay was the fact that when I was working on the reception of Jane Austen in France and Switzerland in the early 2000s, I thought it would be interesting to compare the *incipit* of *Pride and Prejudice* in all translations into French. There are now, in 2022, sixteen French translations of this novel. The *incipit* includes one of the most famous statements in the English language, as well as a modal auxiliary, the rendering of which constitutes a minor challenge for any translator. This essay will analyse all translations of the *incipit*, relating translation choices to historical circumstances, the contemporary status of British literature and attitudes to the translation of fiction as well as to the state of the book market.[1]

There were three early French translations of *Pride and Prejudice*, one of them an abridgement. This compares with one translated into German in 1822, the next one not being published until 1939; and, for Italy, it had to wait until 1932 for its first translation of *Pride and Prejudice*. The speed with which Jane Austen's novels were translated from 1813 onwards may seem remarkable until one becomes aware that this was the case with much British fiction, which was in great demand from the 1740s (as a result of the huge success of Samuel Richardson's *Pamela* [1740–1741] and *Clarissa* [1747–1748]). The French Revolution only increased this demand, as French publishers focused on political works. As for the Napoleonic period, it was unfavourable to literary innovation, because of the strict government control over new publications. This helps to explain why most novels published in France then still focused on sentiment, while new kinds of fiction appeared in Britain—such as the regional novel, by Maria Edgeworth and Elizabeth Hamilton or Sydney Owenson, for instance. The popularity of British fiction in France rose to new heights when Walter Scott turned to the novel in 1814 (Scott 1814) but sentimental fiction remained popular.

After the abridged translation of *Pride and Prejudice* in 1813 (Austen 1813) and the two 1822 translations, (Austen 1822) over a century elapsed before any new ones, no doubt because Walter Scott's novels were hugely popular in France, as they were all over the world, and led to the dominance of social realism in fiction. Then, there was one translation in 1932 (Austen 1932) and no fewer than five in the 1940s alone—three in France, two in Belgium (Austen 1946a, 1946b, 1947, 1948, 1954)—followed by one in 1954 (Austen 1954) and an abridgement for children in 1969 (Austen 1969). There have been five since the year 2000 (Austen 2000, 2001, 2007, 2009, 2011).

For the sake of comparison, one should know that there have been ten translations of *Emma* and eight of *Sense and Sensibility*, the most frequently translated novels after *Pride and Prejudice*. The spate of translations in the 1940s may, I think, be explained by renewed interest on the part of the French and the Belgians in the culture of their allies. Besides, the sale of books translated from English had been forbidden in occupied France (Trunel 2010, p. 226). Further, in 1940, a filmed version of *Pride and Prejudice*, with Laurence Olivier and Greer Garson came out, which may have influenced the choice of publishers to bring out that novel by Austen rather than others. As for very recent translations (some of those based on earlier ones), there is no doubt that they are a response to the many film adaptations of Austen that have been shown in France, both in cinemas and on television, rather than an index to a growing awareness of the cultural importance of Austen's fiction.

Let us first see what has happened to the title of the novel, which, as we know, was not Austen's original choice of title. It is to be noted that *préjugé*, like 'prejudice', can mean both 'damage' and 'preconception'. There is not much variation, the preferred translation being *Orgueil et préjugé(s)*—thirteen times out of sixteen. Recently, there has been a return to the singular *préjugé*, rather than *préjugés*, perhaps because it is felt to be historically more accurate. *Préventions* is a close synonym of *préjugés*, which, however, removes the idea of judgement, and *L'Orgueil et le préjugé* has the unfortunate effect of hypostasising traits of character which are actually shown to become eroded in the course of the plot. The one aberration—and a very entertaining one, too—is *Les cinq Filles de Mrs Bennet*, a choice which may seem wild until one realises that it is a variation on the French title of Louisa Alcott's *Little Women* (1868), *Les Quatre filles du docteur Marsch* [*sic*]. The suggested analogy is, to say the least, misleading! This French translation actually came out in January 1933 (rather than in 1932 as stated on an inside page), and the translators might have known that a filmed version of *Little Women* directed by George Cukor was being shot. In any case, in this title, the moral concerns apparent in the original seem to be replaced by light social comedy.

Now to the *incipit*. I will mostly focus on the opening sentence. A short stylistic analysis of it will make the discussion of it clearer. 'It is a truth universally acknowledged, that a single man in possession of a good fortune, must be in want of a wife.' This is a fairly simple sentence, well-balanced thanks to a first syntagm made up of twelve syllables, a second one of fourteen syllables, and a third one of seven—half as many as the middle and longest one. The two commas, which would have been used in the early nineteenth century to 'punctuate' the syntactic structure of utterances, bring out the ternary rhythm of the sentence. The opening main clause—'It is a truth universally acknowledged'—leads one to expect a verb in the indicative mood: 'that a single man . . . is in want of a wife'. Universal truths are usually declarative sentences; by contrast, here, the verb in the completive clause is, unexpectedly, a modal auxiliary. This auxiliary is highly ambiguous: it may convey obligation or probability, the balance being tilted in favour of obligation by the following sentence in which we see that the surrounding families have settled that the newcomer will marry 'one or other of their daughters'. The second sentence of the novel thus satirically deflates the supposed universality of the first, after the modal auxiliary has already undermined the reliability of the aphorism.

The first translation of *Pride and Prejudice*, which only offers selected excerpts, was published in 1813 over two issues of the *Bibliothèque britannique; ou Recueil Extrait des Ouvrages anglais périodiques et autres* (British Library; or a collection of extracts from English periodical works and other items), a Swiss journal published between 1796 and 1815, specifically aimed at ensuring that francophone Europeans could keep abreast of developments in science and literature in Britain, despite the Revolutionary and Napoleonic wars (Bickerton 1986, passim). It is remarkable that the *Bibliothèque britannique* should have chosen to translate, be it in much-abridged form, a novel by an unknown lady novelist: its editors may have acted on the recommendation of their advisers in London, Alexandre Marcet, a Swiss, and the French bookseller, J.C. Deboffe, for, usually, when translating fiction, the *Bibliothèque britannique* favoured didactic tales, in particular those of Maria

Edgeworth. The translator, who may have been Charles Pictet, one of the editors of the journal, starts pruning the text from the very beginning: 'C'est une verité reconnue, qu'un jeune homme qui a de la fortune doit chercher à se marier' (Austen 1813, p. 373). This first sentence can be translated back into English thus: 'It is an acknowledged truth, that a wealthy young man must endeavour to marry.'[2] The translator seems to have found the word 'universally' redundant, but dropping it entails partly losing the pseudo-portentous tone of the original. 'Must be in want of a wife' is quite well handled, as the translation preserves the ambiguity of the English 'must': the French verb *doit* may convey both obligation or probability.

Of the two 1822 translations of the opening sentence, one is slightly longer: 'C'est une verité presqu'incontestable qu'un jeune homme possesseur d'une grande fortune, doit avoir besoin d'une épouse' ('It is a nearly irrefutable truth that a young man in possession of a large fortune must need a wife'. (Austen 1821, p. 1)) Miss Eloïse Perks (who coyly hides behind the abridged name Mlle E . . . ), being a proper young lady, says that the truth is 'nearly irrefutable'; she also eliminates the ironical portentousness of a 'universal' truth. It is to be noted that the poor standard of printing of this edition is an indication that the book was produced fairly quickly, mostly for the circulating library market.

If I say that the other 1822 translation, which came from Switzerland, is slightly ponderous, I shall be accused of hackneyed prejudice. The fact is that the opening sentence of the novel is rather watered down by being turned into a conditional statement, and by the fact that the 'truth' becomes an 'idea' with no implicit normative value: 'S'il est une idée généralement reçue, c'est qu'un homme fort riche doit penser à se marier', which translates back as 'If there is a widely accepted idea, it is that a very rich man must think of marrying' (Austen 1822, p. 1). The irony and the satirical intention of Austen in her pseudo-solemn beginning are lost; so is the establishment of a discreetly ironical *persona* for the narrator and the subtle blurring of voices she intended in this beginning. Furthermore, the initial *si* selects an idea among many other received ideas, this idea becoming part of the *doxa* rather than being (supposedly, of course) a truth admitted by all human beings.

Significantly, this anonymous translator also leaves out the following sentence from Chapter XXII of Book I: 'Miss Lucas, who accepted him solely from the pure and disinterested desire of an establishment, cared not how soon that establishment were gained (Austen 1988, p. 122).' He or she must have found this sentiment crude, unromantic, and altogether too pragmatic.

The first French translators of Austen would have been used to the narrative voice of Frances Burney, not to the subtle subversiveness of Austen. Hence, the euphemistic phrasing of the opening sentence, especially as *both* 1822 translators may have been women. It seems to me that a much longer and straightforward version (that is, one devoid of irony) of this beginning may be found in Frances Burney's *Camilla* (1796) (Camilla 1796). This novel also opens with generalities on human nature: 'The historian of human life finds less of difficulty and of intricacy to develop, in its accidents and adventures, than the investigator of the human heart in its feelings and its changes', (Camilla 1796, p. 7) which take up one long paragraph. Then there is a second beginning, to Chapter I of Book I: this also takes the form of a general statement, which is not meta-fictional, but prepares the introduction of the characters: 'Repose is not more welcome to the worn and to the aged, to the sick and to the unhappy, than danger, difficulty, and toil to the young and adventurous' (Camilla 1796, pp. 7–8). This, in its turn, is followed by a restricting of the focus to one family: 'In the bosom of her respectable family resided Camilla' (8) (Camilla 1796, p. 8). However, in *Camilla,* the process of moving from the very general to the specific requires about thirty lines, whereas it takes about five in Austen. By the third paragraph, we are in the Bennets' sitting-room.

Apart from translators' familiarity with sentimental fiction of the eighteenth century, which explains some distortions in their translations, publishers at the time usually had low expectations concerning the standard of proficiency of their translators. As far as can be ascertained, any literate person who offered his or her services and produced a coherent

translation was given work. Translation was a job for the hack, the well-rounded young lady, the struggling author, but also, sometimes, the established author, as was the case with Isabelle de Montolieu.

There were no translations of *Pride and Prejudice* in the nineteenth century after the two 1822 ones. This is unsurprising, as Jane Austen's reputation grew very slowly in France, mostly after the publication of James Austen-Leigh's *Memoir* in 1870. However, about every ten years from the 1830s, there were critical essays on Austen's fiction in French periodicals, essays whose authors usually had access to the original English text—having lived in Britain or being teachers of English, for instance. Educated French readers could have bought sets of Austen's novels or individual volumes in English from publishers or bookshops specialising in English-language books, such as Galignani in Paris.

The ebbing of realist fiction in Europe, and the rise of symbolism and modernism contribute to explaining the renewed interest in Austen's fiction in the late nineteenth century. The first twentieth-century translation makes the young man in the opening sentence of *Pride and Prejudice* a bachelor, as in the original; this is the case with all later translations. 'C'est une vérité universellement reconnue qu'un célibataire pourvu d'une belle fortune doit avoir envie de se marier' ('It is a universally acknowledged truth that a bachelor provided with a good fortune must feel like marrying') (Austen 1932, p. 1). It brings in the young man's feelings, which places the matter of marriage in a different sphere from that envisaged by Austen, and makes it difficult to understand 'must be in want of a wife' as submerged free indirect discourse reflecting the thoughts and words of eager parents and young women, for the French phrase now means something like 'must feel like' or even '*must* be made to feel like' if enough emphasis is placed on 'doit'.

Several twentieth-century translations turn the opening sentence into two sentences (Rocart's 1945 translation, Privat's 1946 one, and Vierne's 2001 rendering). Let us first look at Eugène Rocart's translation: 'Un célibataire qui possède une certaine fortune doit absolument se marier. Voilà un axiome universellement admis.' ('A bachelor possessed of some wealth must absolutely marry. This is a universally accepted axiom' (Austen 1945, p. 5).) To state the obvious: the original English sentence was felt to be too long. It is made more colloquial, more 'modern', but the grandiloquence of the original disappears—the seemingly authoritative voice that complacently utters a pseudo-aphorism is more or less erased. There is no commanding pontificator. In the next sentence, Rocart says of the 'man' of Austen's second sentence that he is a 'pauvre garçon' (a 'poor young man'), which explicitly turns the young bachelor into a helpless and unsuspecting victim and alters the stance of the narrator. Shops and Séverac change the point of view, from the positive 'must be in want', to the negative 'cannot do without', thus making the sentence cruder: 'C'est une verité universellement admise qu'un célibataire qui a de la fortune ne peut se passer d'une femme' ('It is a universally accepted truth that a bachelor who has wealth cannot do without a wife') (Austen 1946b, p. 9). This also eliminates the idea of potential pressure exerted by the local gentry. Luce Clarence, in 1954, uses the same phrase as Shops and Séverac. I doubt that Austen would have said that a man could not do without a woman (Austen 1954, p. 7).

In 1946, Privat turns the aphorism into a commonplace: 'Qui songe à en douter? Un célibataire nanti d'une belle fortune doit être nécessairement à la recherche d'une femme' ('Who would doubt it? A bachelor provided with a good fortune must necessarily be in search of a wife') (Austen 1946a, p. 5). The addition of an adverb to the modal auxiliary is redundant and tilts the meaning of 'must' towards obligation or duty, away from probability; the sentence also makes the young man proactive, whereas in the original the need is ascribed to him by the surrounding families about to be mentioned. Privat introduces the theoretical possibility of doubt where Austen parodically embraces the whole world and the deepest truths thereof. Marriage becomes a mundane matter rather than being placed *sub specie aeternitatis*, with the narrator in the role of a great utterer of profound truths. Colloquialism and approximation are creeping into the translation of the opening sentence as if the translator was attempting to reduce the chronological and

cultural gap between the Regency and the post-war period. This may be partly because his translation was published by Editions des Loisirs, which specialised in popular literature. This translation was reissued as recently as 2010.

In 1947, Jules Castier, perhaps because his translation was published in a series, the Bibliothèque anglaise, with a clear focus on major classics, is more accurate than Privat: 'C'est une vérité universellement admise, qu'un célibataire en possession d'une fortune solide doit avoir besoin d'une femme' ('It is a universally recognised truth, that a bachelor in possession of substantial wealth must need a wife') (Austen 1947, p. 2). This translation has a preface by a very famous scholar of English literature, Louis Cazamian, who would have been regarded as a guarantor of authenticity.

Germaine Lalande, in 1948, is, like Privat, over-emphatic: 'C'est une vérité universellement reconnue qu'un riche célibataire doit nécessairement éprouver le besoin de prendre femme' ('It is a truth universally acknowledged that a rich bachelor must necessarily feel the need of taking a wife') (Austen 1948, p. 7). Indeed, in all of the opening section, comprising the first two paragraphs, she is prone to explication and glossing. This was the first illustrated edition of this novel to be published in France; it had forty line drawings, clearly meant to appeal to children.

These 1940s translations have been reissued many times, sometimes with some revision. Austen's second published novel is expected to sell quite well apart from the quality and the accuracy of the translation; indeed, the identity of the translator matters so little that at least two recent reissues do not mention the name of the translator at all! Modern French readers are implicitly assumed to want to read the book of the film, as it were, and not to be concerned with getting as close as possible to the linguistic sophistication, the subtleties and ambiguities of the original. The next translation, by Gilberte Sollacaro, was a digest published in 1969 by Sélection du Reader's Digest, a publisher specialising in 'potted' culture. This abridgement was offered in a single volume with two other works for young people, one of them *Les Aventures de Sherlock Holmes* (a collection of stories by Conan Doyle). The salient fact is that Austen was considered famous enough to be issued in a collection for young people—not specifically young girls. This edition has sixteen coloured prints by the well-known British illustrator Charles Brock, taken from a late nineteenth-century British edition of *Pride and Prejudice*. The opening sentence is not condensed and reads: 'Tout célibataire en possession d'une fortune solide doit nécessairement—et c'est là une vérité universellement reconnue—être en quête d'une épouse' ('Any bachelor in possession of substantial wealth must be—and that is a universally recognised truth—in quest of a wife') (Austen 1969, p. 316). Fact is foregrounded to the detriment of the endorsement by an omniscient voice of a general truth, and the young man is presented as more active and anxious to find a mate than Austen depicts him. Further, 'any' bachelor is more emphatic and implicitly quantitative than 'a' bachelor or young man.

In 2000, Jane Austen finally appeared in the prestigious leather-bound Pléiade series of Gallimard, being then, with the Brontë sisters, the only British female author to be granted her own volume. Jean-Paul Pichardie has a fairly slapdash approach to the text: 'Il est universellement admis qu'un célibataire nanti d'une belle fortune a forcément besoin d'une épouse' ('It is universally accepted that a bachelor provided with a good fortune necessarily needs a wife') (Austen 2000, p. 561). The 'normative' truth disappears, hence also the submerged *persona* of the pontificating narrator, and 'forcément' is much too colloquial, besides being anachronistic. A year later, Béatrice Vierne chose to say: 'Il est une vérité universellement admise: c'est qu'un célibataire doté d'une solide fortune a certainement besoin d'une épouse' ('There is a universally accepted truth: it is that a bachelor provided with substantial wealth must most probably need a wife') (Austen 2001, p. 13). She opts for two independent clauses, with an anaphorical link, which is close to the subordination in the original sentence but leaves out the seme of obligation. This edition clearly aimed at capitalising on the success of the 1995 BBC Series, as its front cover displayed a still from it. Pierre Goubert, in his 2007 rendition, introduces a human agent at the beginning of the sentence, an embodiment and a particularisation which eliminate the hyperbolic

pseudo-universality of the statement: 'Chacun se trouvera d'accord pour reconnaître qu'un célibataire en possession d'une belle fortune doit éprouver le besoin de prendre femme' ('Everybody will agree that a bachelor in possession of a good fortune must be in want of a wife') (Austen 2007, p. 35). In 2009, Laurent Bury eliminates the ambiguity of 'must be' by saying, like some of his predecessors, that the bachelor 'is necessarily looking for a wife' ('est nécessairement à la recherche d'une épouse'), (Austen 2009, p. 7) thus, again, making the young man active whereas Austen represents him as a passive being whom eager mothers of marriageable young women hope to make a prey of. Finally, in 2011, Sophie Chiari re-introduces the modal 'must' but also has the young man look for a wife, if in a slightly higher linguistic register ('doit être en quête d'une épouse') (Austen 2011, p. 35).

As can be seen, there is an amazing variety of translation choices for such a relatively short sentence. The translation of 'must' shows that Austen's polysemy and tendency to understatement eludes some translators, or at least they do not bother to account for those characteristics accurately. More subtly, and this may not be the exclusive responsibility of the translators, the original layout, with the opening statement being set off, is only respected in a little over half of the translations. Again, this partly reduces Austen's mock-solemnity. It is notable that the latest translators have not tried harder than their predecessors to be accurate, to account for the carefully balanced rhythm of Austen's first sentence, and the tension between universality and specificity.

Few translators of Austen have discussed their work or given any assessment of Austen's fiction. Testimonies are particularly valuable for the earlier period, when Jane Austen was not yet regarded as one of Britain's greatest writers and when translators had a 'naïve' approach to the English original. We are fortunate that there are a few lines from Eloïse Perks, whose translation of *Pride and Prejudice* was published in 1822, as we saw. Her short preface is available in some copies of the first volume. She begins with the usual exercise in *captatio benevolentiae*, felt to be all the more necessary as she is not just a woman, but a young one; in the same breath, she explains that Austen is noteworthy because she has been translated by the famous Isabelle de Montolieu—her *Raison et sensibilité, ou les deux manières d'aimer*, a very loose translation of *Sense and Sensibility*, had been published in 1815, benefiting from the fame of the translator:

> It is no doubt bold on the part of a young stranger, who is still new to the art of writing, to dare and translate an author that the elegant pen of Mme de Montolieu has made so well-known; but is it not better to imitate—to the extent that that is possible—a good model rather than a mediocre one? (Gilson 1997, p. 142)

> Next, she focuses on the novel itself:

> The novel entitled *Pride and Prejudice*, which I am now bringing before the public was published, in England, shortly after *Sense and Sensibility*, the translation of which has been so successful in France. What makes this work worthwhile is not the plot but a detailed picture of manners painted by a well-bred woman. I have ventured to include explanatory notes about aspects of our manners which, though they are those of a neighbouring country, are but little known in France; if the reader approves of this care, if it means that my translation will be read with some pleasure, I will have reached my goal, and I will feel encouraged to publish the other works of the same writer which have not yet been translated. (Gilson 1997, p. 142)

We may compare this with Alexandre-Nicolas Pigoreau's comment in his *Premier Supplément à la petite bibliographie biographico-romancière, ou Dictionnaire des romanciers* (First Supplement to the short biographic-romantic Bibliography), which came out in December 1821, just after *Orgueil et prévention* was issued, though the front pages of both works bear the date of 1822—as was standard in Britain too, books were dated the following year when they came out very late in the year. This is what he writes in what is essentially a publisher's catalogue:

A small group of people is enough to Jane Austen to paint a picture of English manners; a picture in which many a French reader will recognise himself. This work is not for those who look for a lot of action and rush to the dénouement; one must attend to details and to the art with which the author has each of her characters act and speak according to the personality she has given them. The translation is by Miss Eloïse Perks, a young Englishwoman brought up in London; her notes wonderfully highlight the talent of the author. (Pigoreau 1821, p. 8)

Both Eloïse Perks and Pigoreau lay emphasis on *Pride and Prejudice* as a picture of manners. Pigoreau is more perceptive than Perks when he notices that each character has an idiolect and behaves in a way consistent with his or her personality. Both commentators also remark on the importance of detail in Austen's fiction; this is akin to what Walter Scott, in his insightful review of *Emma* (published in the *Quarterly Review* in 1816) calls 'the merits of the Flemish school of painting' (Scott 1998, p. 295). They are struck by the introduction of a new kind of realism, though, of course, they do not use that word. This cultural distance explains why Eloïse Perks provided some notes covering British manners, places and foods.

This perception of Austen as primarily a novelist of manners—a perception which differs from that of the early British reviewers of *Pride and Prejudice*, who pay particular attention to the handling of character—is confirmed by the sub-title of the French translation of *Emma*: *La nouvelle Emma, ou les caractères*[3] *anglais du siècle* (*The New Emma, or Modern English Manners*). While Pigoreau implies that Austen's 'pictures' have a universal scope, by which I think he means a moral scope, he, and Eloïse Perks even more clearly, are alert to the Englishness of Austen's fiction. This gives it its anthropological interest. This relative strangeness probably explains why *Pride and Prejudice*, a novel with so many *peripetia*, may have been received as essentially a novel of manners. One should remember that the gentry, which Austen focuses on, had no equivalent in French society: this in itself was enough for Perks and Pigoreau to respond primarily to the account of mores and to social interaction in *Pride and Prejudice*.

In the absence of any reviews of the early translations of this novel, these are the only two assessments we have. There being no reviews is unsurprising: French periodicals generally carried far fewer reviews of fiction than their British equivalents: the *Mercure étranger* (The Foreign Mercury, 1813–1815) had no time for fiction at all; the *Quinzaine littéraire* (The Literary Fortnight, 1817–1818) only carried three reviews of British fiction in 1817, three novels all of which were translated in that year: Lady Morgan's *Novice of Saint Dominick* (1806), Jane Porter's *Pastor's Fireside* (1817) and Maria Edgeworth's *Ormond* (1817). Things changed with the arrival of Walter Scott on the scene (from 1816 in France with the translation of *Old Mortality*), but there was no good reason for periodicals to take note of anonymous (and seemingly) sentimental novels such as Austen's. Tellingly, as has been said, the first translation of *Sense and Sensibility*, *Raison et sensibilité, ou les deux manières d'aimer* was a great success primarily because it was by a famous novelist, Madame de Montolieu, who 'naturalised' this novel by turning Austen's critique of sensibility into a eulogy of it! It became hers so much that it was included in her complete works.

The two assessments of *Pride and Prejudice* available for the 1820s are valuable because they implicitly perceive that Austen is distancing herself from the sentimental novel which had been the dominant form of fiction for decades in Britain.

However, critics and translators alike thought that the Englishness of Austen to some extent had to be domesticated, especially in matters of style. This is what Pigoreau wrote (in 1821): 'French vivacity is unsympathetic to British phlegm, which dwells on any idea whatever and presents it under a thousand different aspects.' (Pigoreau 1821, p. 12) Unsurprisingly, from a French point of view, this phlegm makes for long-windedness and much pointless detail. Pigoreau must, however, be credited with adding that 'everybody must keep his national manner; an Englishman in French costume is graceless'.

Like Pigoreau, some translators still thought, as did Abbé Prévost (1784) when he translated Richardson's novels, that English fiction had to be made stylistically crisper,

leaner, and more vivacious. Here is a brief excerpt from Prévost's views on the matter, as stated in his preface to the *Nouvelles Lettres angloises ou Histoire du Chevalier Grandisson* [*sic*] (New English Letters or History of Sir Charles Grandisson), his translation of Samuel Richardson's last novel, published between 1755 and 1758:

> Without altering the author's general design or even the greater part of its execution, I have given his book a new aspect by cutting out wearisome excursions, overwrought descriptions, unnecessary conversations and irrelevant reflections. (Cointre and Rivara 2006, p. 71)

Like many readers, he found *Sir Charles Grandison* somewhat pompous and long-winded. In the introduction to his 1751 translation of Richardson's *Clarissa*, he had already affirmed the extensive rights of a translator:

> By the supreme right of any writer who aims at pleasing in his native language I have changed or suppressed what was not in keeping with that goal. I am not afraid of being charged with excessive rigour. English literature has been known in Paris for twenty years now and it is generally admitted that, to be naturalised, it often requires such small repairs. But I have considered it a duty to keep the national air of characters and customs. A translator's rights do not extend to altering the substance of a book by dressing it in a new language. In any case, is it necessary? A foreign air is rather an asset in France. (Cointre and Rivara 2006, p. 67)

Prévost could be so categorical because the domestication of foreign texts, their integration into the target culture—in this instance, French culture—were still seen as a way of adjusting them to a universal good taste that was better served in one's own country. Anglomania in France consisted of absorbing otherness rather than acknowledging it; the same applied to the 'Englishing' of French literature. As this suggests, there was some contradiction between the demand for foreign literature and the urge of translators to adjust it to the supposedly higher standards of the native culture. Of course, this was not perceived by translators in general, and Prévost in particular, as contradictory, as the domestication of foreign texts was thought to be a way of making them universal.

So, there was some kind of 'theoretical' justification for, usually, pruning, and sometimes for expanding, the text of a novel, though the turn of the nineteenth century saw a shift towards trying to convey strangeness and foreignness rather than domesticating it. Isabelle de Montolieu was unusual in altering the plot of *Sense and Sensibility* in 1815—but that was because she was so steeped in the conventions of the novel of sensibility that Willoughby *had* to marry Eliza Williams (called Caroline in her translation). As a general rule, the progressive and sometimes incomplete switch from a remedial domesticating approach in translation to a foreignising and contextualising attitude took place in the late eighteenth and early nineteenth centuries. Emblematically, in the advertisement for the 1814 translation of Jane Porter's *The Scottish Chiefs* (1810), the translator gives two long quotations from David Hume's *History of England*, to provide information about William Wallace, the protagonist of the novel. In the last paragraph, he still engages in the standard exercise in *captatio benevolentiae* for his translating work, but also evinces interest in the dynamics of narrative and the importance of details to the integrity of the work:

> We will only say that we have been faithful to the text, without turning respect for our author into an obsession. We never made free to impose our own notions on him; we have merely pruned a few *longueurs*. We may have been remiss in being too restrained there; but, by unthinkingly removing whole passages, we might have run the risk of suppressing details material to what followed or of undermining effects which those details were meant to prepare for. (Cointre and Rivara 2006, p. 263)

These programmatic words are closely echoed by Pigoreau in his assessment of *Pride and Prejudice*, a few years later. With the development of the historical novel, the source-text became not simply where translation begins, but also an important historical source for

the target-text with which it ends—hence the higher standards of accuracy of translations of Scott's novels and of nineteenth-century literature generally. The improvement in the status of the novel brought about by Walter Scott, with fiction finally acquiring intellectual respectability, may have led to more accuracy in the translation of Austen and more attention to details of all kinds.

However, into the late twentieth century, the 'pruning' approach still had some followers. Though it does not show very much from the *incipit* of *Pride and Prejudice*, most translators have indulged in leaving out words or phrases when translating Austen's texts. More drastically, in the twentieth century, two abridgements of *Pride and Prejudice* were presented as translations; Germaine Lalande (1948) and Luce Clarence (1954) quietly abridged the text but this was not mentioned anywhere in the books. The former translation was published in a series of books for children, but one would still have expected some acknowledgement of abridgement.

That by the twentieth century critics expected more accuracy in translation is evinced by F. Delatte's short but scathing review of the 1932 translation of *Pride and Prejudice* by Leconte and Pressoir. The two women mostly translated children's books: clearly, the editor at the reputable Plon publishing house saw Jane Austen as very much a 'girl's author'. To Delatte, a man who was presumably a teacher, the age of the 'free adaptation' by amateur lady translators was over. Here is the bulk of what he says in his essay 'La Traduction de *Pride and Prejudice* de Jane Austen' published in the *Revue de l'enseignement des langues vivantes* in 1934. He lists seven types of failings he came across, each time giving two examples: '1. Whole paragraphs go untranslated. 2. Whole sentences are left out. 3. Some phrases are left out. 4. Translation is sometimes vague or inaccurate. 5. The text is sometimes merely paraphrased. 6. The specific meaning of some words is weakened. 7. There are over-emphases'.[4] Delatte concludes that the vaunted translation is actually no more than an adaptation. Though this is certainly a very poor translation, it was re-issued as recently as 2020, no doubt still to exploit the vogue of film adaptations. Rather than as a master of satire and irony, Austen is still marketed in France as a writer of delightful sentimental comedies taking place in the equally delightful English countryside.

To conclude. For my chapters for *The Reception of Jane Austen in Europe*, I examined a great many translated passages from all the novels, comparing them in detail with the original.[5] It may seem difficult to believe but, with very few exceptions, French translators have not done justice to Austen's style, to the complexities, the subtle nuances, the shimmering of her writing. They have not given Austen the time she requires. Painstaking translation, in this case, would do no more than mirror the painstaking writing process. The first sentence of *Pride and Prejudice*, with its hiatus, namely, the incompatibility between announcing a truth calling for the indicative mood and ending the sentence with free indirect discourse reflecting the thoughts of individuals, relies on the polysemy of 'must', which also exists in French, though some translators have ignored or not perceived it. More challengingly, this *incipit* resorts to free indirect discourse, of which Austen was one of the first practitioners in fiction. Free indirect discourse was then and still is much more common in English than in French, though Gustave Flaubert was a great master of it. Some French translators felt the alienness of this grammatical form so much that, when translating Austen, they turned it into direct speech (this is fairly systematic in Montolieu's translation of *Sense and Sensibility*). This blurring of free indirect discourse and thus of the narrator's voice in the incipit of *Pride and Prejudice* and throughout early French versions of Austen, also explains why so much of her irony disappeared in translation. This is another reason why Austen was perceived as a novelist of manners: when the irony is diluted, the representation of occurrences in the life of a few gentry families comes to the fore. The challenges presented by the translation of Austen's fiction are often circumvented because most translators of Austen in the twentieth and the twenty-first centuries have been professionals needing to make a living (or academics with many different calls on their time), who cannot afford to spend half a day over one page. In this respect, the translation of the first sentence of Austen's second published novel is emblematic: many are the translators who have

stumbled over its 'must' and, indeed, over the sentence as a whole. So, access to Austen's very idiosyncratic narrative voice is still limited and the reception of Jane Austen in France is, in a sense, still ahead of us.

**Funding:** This research received no external funding.

**Informed Consent Statement:** Not applicable.

**Data Availability Statement:** Not applicable.

**Conflicts of Interest:** No conflict of interest.

## Notes

[1] On translations of Jane Austen into French, see (Cossy 2006; Trunel 2010; Mandal and Southam 2007; Trim 2015).

[2] Translations are the author's own unless stated.

[3] The word 'caractères' is taken in a very specific sense here, which was revived at the beginning of the nineteenth century in France; some of the semes it conveys have to do with moral qualities, others with 'characteristics'; it is not the equivalent of the English 'characters' in the sense of 'literary characters'. It is a meaning which comes from Theophrastus and La Bruyère. It is close to the English *type* or *constitution*.

[4] (Droxler 1934). Here are the two examples he selects: 'How can you *abuse* your own children in such a way" = 'parler ainsi de ses propres filles' [to talk of one's own daughters in this way]'; 'a common failing' = 'un sentiment très répandu' ['a common sentiment]' (214).

[5] Isabelle Bour, 'The Reception of Jane Austen's Novels in France and Switzerland: The Early Years 1813–1828'; 'The Reception of Jane Austen's Novels in France and Switzerland: The Later Nineteenth Century, 1830–1900' and 'The Reception of Jane Austen's Novels in France and Switzerland: The Modern Period, 1901–2004: Recognition at Last?' in (Mandal and Southam 2007, pp. 12–73).

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
