# Peer review of "What Happened to the ‘Truth Universally Acknowledged’? Translation as Reception of Jane Austen in France"

_humanities, doi:10.3390/h11040077_

Round 1

Reviewer 1 Report

This essay provides a lively and illuminating account of Jane Austen's reception in francophone Europe by way of examining the titles and first sentences of the extant canon of translations. English readers with a knowledge of French will be delighted by the many ways that the first sentence was translated and the author's excellent re-translations into English, which beautifully capture the nuances of the French versions and their differences from Austen's original. The essay is mostly of value for its overview of French sources and close reading of individual translations, as well as for its account of general differences between English and French culture (literary, journalistic, translational, social) in the first half of the nineteenth century. I also enjoyed the contextual details behind individual translations that appeared in the wake of (for example) films. There are some other speculations about the novel of manners, sensibility, and gender that are suggestive but do not rise to the level of robust argument, and on these topics there is little engagement with other scholarly sources. 

I was particularly intrigued by the comment at the end that French literature is comparatively bereft of free indirect style (Flaubert comes to mind as a counter-example); perhaps the author meant that during the early years of Austen's reception, free indirect style had not yet taken hold as a technique? Or is it really the case that French authors throughout the nineteenth and twentieth centuries do not employ this mainstay of English prose? It would certainly offer a compelling explanation for why so many translators miss this aspect of Austen's incipit, but without greater knowledge of French literature I can't defend the justice of this claim, which would benefit from further substantiation and/or development.

A few other stray comments and questions:

1 why “potential novelists” and not “potential readers”? Are you suggesting that translations were aimed at writers rather than readers?

2 it's not clear why popularity of Scott’s novels would prevent new translations of Austen

3 I'm not sure it’s fair to say that adding the definite article has the effect of hypostasizing traits in a way that departs from the original – in English, pride and prejudice can refer to abstract universals as much as the pride and prejudice of the novel's individual charactres

4 the relevance of this statement is unclear: “It is to be noted that the poor standard of printing of this edition is indication that the book was produced fairly quickly, mostly for the circulating library market”

4 it's not clear to me why female translators would be more inclined to euphemize than male ones. On the contrary I would expect men to go farther in order to protect the dignity/virtue or women

5 I don’t see why this pretty accurate translation is “anachronistic”

6 this is vague: “probably for reasons having to do with copyright laws." What reasons? why?

7 “actually a collection of stories by Conan Doyle” – don’t see why you need the “actually” – did French readers not know that Doyle was the author? Also Arthur Conan Doyle is the complete name, and Doyle is the accepted surname. 

8 italicize Orgueil et prévention

8 “as rather a static novel” – should read “as a rather static novel,” but the inference may not bejustified. Many in the period compared novels to pictures or portraits without implying that they lacked motion or dynamism.

9 this is over-general and adds little – recommend to delete: "(including Gothic fic-
tion which offers variations on the novel of sensibility), but which the British were moving away from, especially with regional and historical fiction, whereas in France authors still wrote comfortably in that mode, as has been said.”

9 missing closing quotation mark on Pigoreau quote

Author Response

I have taken into account the suggestions of both reviewers, deleting a few sentences and adding a few, the latter being highlighted in green
I have added italics where they were missing, also highlighting them in green
I have corrected a very few typos and added first names for the sake of consistency
I have quietly removed a few unnecessary spaces and added a few commas
I have highlighted in red a quotation where the spacing seems to be wrong (pages 9-10)

Thank the two reviewers for their generous and very helpful comments.

Reviewer 2 Report

This is a comprehensive and enjoyable account of how Austen's most celebrated novel has fared in France since its first (abridged) translation appeared in 1813. 

Reference should somewhere be made (perhaps in the first footnote) to Gillian Dow's articles and essays on Austen and French translation. For comparative purposes it would be useful to know if the same pattern in French translations that is identified on p. 2 applies to translations of Austen into other languages (I *think* it is broadly similar in German, for instance: a couple of early translations, then nothing for about a century and a resurgence of interest in the 1930s or 40s?). With reference to the title of the novel (pp. 2-3), fuller discussion of the meanings of 'Prejudice' as opposed to 'préjugé(s)' would be very welcome, as the distinction between the two suggests another interpretative crux before we even get to the first sentence -- it should perhaps also be noted, briefly, that this was not Austen's original choice of title.

Once the essay moves on to present and evaluate the details of different translations of the first sentence the discussion is for the most part assured and engaging, offering useful contextual details regarding the market for fiction, the aptitudes of various translators, and the history of Austen's reception in France. It is mysterious that French versions of Austen have fallen short in translating her free indirect style into a plausible and subtle native idiom, since in Flaubert they had a home-grown equivalent of comparable brilliance in that complex yet unassuming mode.

I am not sure that 'euphemisation' (p. 4) is a word -- or, if it is, that its usage should be encouraged.

Author Response

(The authors gave the same response as above.)
